# The Effects of Dust Storms on People Living in Beijing: A Qualitative Study

**DOI:** 10.3390/ijerph21070835

**Published:** 2024-06-26

**Authors:** Zhaohe Chang, Susan Bodnar

**Affiliations:** Teachers College, Columbia University, New York, NY 10027, USA; sb2581@tc.columbia.edu

**Keywords:** dust storm, Beijing, environmental impact, psychological well-being

## Abstract

Dust storms, which are common aversive occurrences in northern China, result from high winds, dry soil or dust, and soil surface disturbance. Exposure to dust storms, regardless of duration, can induce varying mental and physical distress levels. Recognizing the urgency of comprehending the impact of dust storms on residents and the scarcity of information on their effects on the indigenous civilians there, this study aims to address this gap by qualitatively sampling 29 participants from Beijing, a typical city in northern China. The current study seeks to gain insights into residents’ dust storm experiences and explore their perspectives on effective coping mechanisms. The findings align with existing knowledge regarding the mental and physical repercussions of dust storms while identifying some emerging patterns of coping mechanisms already employed by residents in Beijing. Concerns regarding mental well-being, either directly influenced by the environmental conditions or indirectly stemming from disruptions to life routines on a broader scale, persistently dominate people’s perceptions of dust storms. New themes emerged following the step-by-step exploration of feelings and coping mechanisms. This study aims to enlighten the public about the ramifications of the dust storms in Beijing and advocate for essential policy support.

## 1. Introduction

Aversive environments, characterized by unpleasant, stressful, or uncomfortable conditions, can significantly impact mental health. Numerous studies [1,2,3] have established a link between exposure to such environments and mental health issues like anxiety, depression, and post-traumatic stress disorder (PTSD). One prominent mechanism through which aversive environments affect mental health is stress. Prolonged exposure to stressful conditions can dysregulate the hypothalamic–pituitary–adrenal (HPA) axis, which is part of the body’s stress response, leading to increased cortisol levels and contributing to mental health problems [4]. Exposure to aversive environments can also induce feelings of helplessness and a lack of control, further exacerbating stress and anxiety [5].

Dust storms, which are typical aversive events in northern China, result from high winds, dry soil or dust, and soil surface disturbance. One readily observable impact caused by dust storms is polluted air and its detrimental effect on human physical health, primarily attributed to the dust particles’ biological and chemical properties. A recent nationwide study conducted in the US, derived from the life satisfaction (LS) data from the CDC Behavioral Risk Factor Surveillance System (BRFSS) over the past five years, found that individual daily life satisfaction went down by 0.008 during the occurrence of a dust storm due to the sharp reduction in local air quality [6].

The increased ambient particulate matter (PM) usually comes with significant loads of microorganisms and toxic biogenic allergens, which are accumulated in the source areas, trapped in the air, and eventually transported to the affected area [7]. These short-term health effects include respiratory, eye, and skin issues, while long-term exposure can lead to chronic respiratory diseases such as bronchitis and emphysema, as well as respiratory illnesses like silicosis (also known as dust lung) in extreme cases [8]. Besides the duration of exposure to the dust storm, the particle size of dust storms has also become a significant concern of many health professionals, as some preliminary associations are being found between the size of particles and the manner of illness. Exposure to dust could significantly boost mortality rates and hospital visits due to respiratory and cardiovascular diseases [8,9]. Some other effects of dust storms could entail traffic accidents and disruptions to daily life routines.

While the direct impact of dust storms on mental well-being has not been thoroughly investigated, the broader psychological effects of climate change can serve as a model for this question. A review of 13 studies [10] found a link between air pollution exposure and poorer mental health outcomes, including dramatically exaggerated risks of depression and anxiety potentially implicated by dust storms. Another central concept closely related to climate change’s cognitive impact is Solastalgia. Solastalgia, introduced by Glenn Albrecht, describes the distress caused by negative perceptions of one’s home environment. More specifically, it marks the feeling of “pain caused by loss of solace connected to the present state of one’s home” [11].

Another impact of dust storms on individuals is the enforced stay-at-home situation during their occurrence [12]. While having a known negative impact on well-being, self-isolation lacks consistent agreement on the nature of its effects. Individuals with pre-existing health conditions are particularly vulnerable to the negative impacts, especially mental health symptoms, during self-isolation or confinement, as evident from the COVID-19 pandemic [13]. However, the psychological effects of self-isolation and confinement extend beyond their direct impact, with many circumstantial implications potentially causing profound impacts on health.

Compelled self-isolation can result in a loss of relatedness to both nature and social connections, both of which can adversely influence psychological well-being and induce varying levels of psychological distress [13,14,15]. Similar to the well-established positive association between the presence of nature in the living environment and self-reported health, the absence or loss of this connection could be detrimental to one’s mental wellness [15]. Additionally, the effects of disrupted social ties during self-confinement must be addressed. A healthy and supportive social network benefits psychological well-being, according to the main effect model or the stress buffering model. Given the lack of consensus on gender differences in the benefits derived from maintaining social ties, the current study sought to address this question by incorporating a list of demographic inquiries.

Moreover, northern China has seen an increasing frequency of dust storms in recent years [16]. After nearly a decade of peace, severe dust storms recurred in 2021 and have continued ever since then, driven by a strong Mongolian cyclone that transported large amounts of dust particles into northern China [17]. For example, the dust storm that persisted for two weeks and swept across Mongolia in the spring of 2021 impacted 8000 people and 2000 households across 14 provinces, profoundly extending to most of East Asia. As a result, the most immense dust storm ever recorded in the past ten years occurred in Beijing, with its outcomes daunting and significant enough for another country somehow distant from the affected area, South Korea, to issue their warning of “yellow dust” for the first time in a decade. 

This signals a critical time, emphasizing the urgency of studying dust storms in northern China and some commonly affected areas like Beijing. To better capture the direct mental impact brought by the dust storms on local residents and to prepare for future events, this study aims to provide a comprehensive understanding of the physical and psychological effects on indigenous people in Beijing during past dust storms. Moreover, this study expects to gather insights on measures necessary for sustaining their well-being to bridge some existing gaps in the current academia.

## 2. Materials and Methods

The present study adopted a qualitative research design, utilizing an online survey created through Qualtrics to collect participant data. The survey comprised primarily qualitative questions and a list of demographic inquiries to ensure a comprehensive exploration. Seven qualitative questions were incorporated, with examples such as “In what ways do dust storms impact your daily life and routines?” and “Can you recount specific incidents or situations related to dust storms that may have affected your mental well-being?” (see Appendix A). Additionally, demographic questions, including age, gender, and duration of residence in the city, were also included.

Initially developed in English, a bilingual survey was created as the sample was limited to individuals residing in Beijing. Back translation was conducted to ensure translation accuracy and efficiency. The sample consisted of Beijing residents who had experienced dust storms and were still residing in the city. The anticipated sample size was approximately 20 to 30 participants aged 18 and above. Participants were recruited through snowball sampling via the widely used Chinese social media platform WeChat, which boasts over 1.2 billion global users, predominantly in China [18]. Snowball sampling was selected as the primary recruiting method for qualitative analysis due to its convenience and flexibility. It was particularly advantageous for assessing a hard-to-reach sample who might be geographically dispersed, unrecorded, and/or desiring anonymity [19]. Given the present study was only interested in residents living in Beijing who had experienced dust storms, the use of snowball sampling was legitimate. Thematic analysis was employed to identify patterns and themes in the data. Demographic information, including age and gender, along with all responses, was collected and securely stored following the Institutional Review Board (IRB) guidelines of Teachers College, Columbia University. The IRB protocol for the current study was approved on 17 January 2024, and the protocol ID is 24-170. 

### Participants

The current study incorporated 29 participants, ranging in age from 24 to 63. To align with the study’s objectives, participants were exclusively selected if they were above 18 years of age. Among these participants, excluding two who chose not to disclose their gender identity, 14 were male, resulting in a well-balanced gender representation. A crucial criterion for inclusion in the study was a minimum residence of one year in Beijing, with participants’ duration of residence spanning from 4 to 63 years. Individuals not meeting the age or residence duration conditions were excluded from the study.

The participants’ occupational information was also gathered, revealing a diverse spectrum of professions. Examples include medical professionals, teachers, athletic coaches, students, engineers, retirees, and freelancers such as cosmeticians and tea art specialists. This broad range of occupations ensures the survey’s extensive coverage and the inclusion of diverse voices, enhancing the generalizability of the study results. A preliminary inquiry into participants’ preexisting health conditions indicated that 25 had no preexisting health conditions.

The current study employed thematic content analysis. This approach began with extracting codes from individual responses for each question. These codes, sharing similar meanings or characteristics, were amalgamated to form more extensive themes and topics [20]. 

## 3. Results

The average study completion time was approximately 10 min. Among all participants, 25 out of 29 (86.2%) reported experiencing some emotional change during dust storms. Beyond the impact on their mental well-being, many participants expressed concerns about difficulties in their daily lives and transportation caused by dust storms. A more detailed systematic analysis of written responses will be provided below in the order of the survey questions.

### 3.1. Recent Experiences of Dust Storm

The survey commenced by collecting participants’ general impressions of a recent dust storm in Beijing. This question gathered overall feelings or thoughts people may have without indicating a specific area. Results showed that 13 out of 29 participants shared experiences of a severe dust storm during the summer of 2023. This present finding corresponds with the ascending trend of dust storm frequency in China, reported earlier this year as a reflection of climate change on the Mongolian plateau. However, four responses suggested a less severe recent experience, mentioning statements like “there haven’t been any dust storms recently” and “recent experiences living in Beijing are better compared with the smoggy conditions in the past”. This improvement may imply governmental policy changes and technological advancements, which will be discussed later in the passage.

Responses about recent dust storm experiences can be categorized into themes: perceptual, emotional, and realistic difficulties related to transportation and daily life. Perceptual indicators are further divided into visual, olfactory, and holistic perceptions of pollution conditions. Visual descriptions include statements like 

“*The entire sky is in a dusty yellow color, and the visibility is very low” and “the description conveys a sense of the sky becoming dim. The once bright sunlight is obscured, casting an overall hazy grayness across the sky. If one were to stand at an elevated position and look around, they would observe that distant mountains, buildings, and other elements are veiled, visible only as blurry outlines*”.

Furthermore, numerous participants noted their olfactory perceptions during the dust storm, with typical descriptions including sentiments like “it feels like there’s a taste of dust in the air” and expressions like “I can’t breathe!” Two participants unanimously employed the metaphor “end of the world” to articulate their feelings during the dust storm, emphasizing the haunting nature of the event in Beijing. All these instances directly illustrate people’s emotions during the dust storm conditions.

The last theme from recent dust storm impressions revolves around the impact on daily life and transportation. Examples include “Eyes easily catch dust, and the face gets dirty as well” and “Going outside, one ends up with a face covered in dust, resembling someone who has been doing farm work for years”. The realistic impact and maladaptive sensory perceptions contribute to another major theme: the effects on mental well-being. This mental impact can be summarized into two nodes: “depressing” and “shocking”. Further analysis of this mental implication will be presented later in the passage.

### 3.2. Dust Storm Impacts on Daily Life and Routines

Following the general inquiry about participants’ impressions of the dust storm, the second question aims to investigate the dust storm’s impact on people’s daily lives and routines. Commencing with this inquiry, each subsequent question in the survey is tailored to explore a specific domain of impact that follows the study’s interest. This approach ensures a focused examination of various facets related to the subject matter, allowing for a nuanced exploration of the diverse dimensions under consideration. Responses to this question reveal two major themes: difficulties caused by living conditions and transportation (physical impact) and subsequent concerns about physical and mental wellness.

To begin with, the majority of participants shared their unpleasant experiences during the dust storm. One of the most prevalent words in their responses is “inconvenience”, which could be further divided into two separate categories according to the outcome. The most direct impact of the dust storm on people’s lives is significantly reduced visibility. Examples of responses include “It affects travel; the visibility is too low for commuting by car” and “Reduced visibility poses a hazard to transportation safety”, which could escalate into more significant mental concerns if transportation safety is not guaranteed. Another aspect of inconvenience arises from the direct impact of dust on life routines. Due to the excessive amount of dust brought by the storm, house cleaning becomes a significant challenge. For instance, some participants shared that during the dust storm, “there is more dust at home, requiring more frequent car washing than before” and “the skin feels dry, and there’s a strong earthy smell, making it impossible to open windows”.

Beyond the direct impact on daily life, additional implications involve challenges to travel plans, which will be explored in subsequent sections. Due to the excessive dust and strong winds, many individuals reported having to make necessary adjustments to their travel plans. For instance, eight participants highlighted difficulties in their travel plans, such as being unable to go out as freely as they normally would during the dust storm compared to other days. Other significant behavioral changes emerged, including the necessity to wear masks during dust storms and the avoidance of going out altogether. Notably, reasons for staying indoors varied and extended beyond inconvenience, encompassing concerns for well-being related to the extreme weather conditions, warranting a meticulous examination and discussion of the mental implications.

A proportionate number of participants shared that they refused to go out because of potential concerns for their physical health. This concern mainly stems from the uncertainty they feel about the outcomes, with remarkable attention paid to their well-being and that of other family members, representing the second central theme identified. One participant illustrated this concern: “Due to concerns about health, reduced outdoor activities, spent two days indoors with an air purifier running”. Specifically, 13 out of 19 participants expressed concerns about the potential harm caused by the dust storm to physical well-being, with respiratory problems being a primary concern. Examples of responses include 

“*Indeed, due to the high levels of dust and particulate matter in dust storm weather, its impact on human health can be significant;” “Difficulty in breathing is particularly noticeable in such conditions;” and “Having nasal inflammation already, the dust storm exacerbates the sensation of a completely blocked nose*”.

Notably, these major health concerns during dust storms contribute to widespread mental instability. One significant impact on mental well-being is fatigue or dizziness, partially attributed to self-isolation and reduced outdoor activities. Additionally, two more participants reported feeling a lack of interest during dust storms. The specific reasons behind these reported symptoms due to confinement could be multifactorial; therefore, future research is warranted.

### 3.3. Changes in Mood or Emotional Well-Being during Dust Storms

The subsequent questions delve into a detailed exploration of the impact on mental well-being during a dust storm. Eight participants expressed that their experiences during the dust storm were generally unpleasant, signifying a shared sentiment regarding their mental state during such events. The effects on mental wellness can be categorized into three primary themes: those caused by the weather, past experiences, and concerns about personal or others’ health outcomes.

The primary theme identified in the responses pertains to the mental impact caused by the weather itself. This includes feelings of boredom, irritation, depression, and, in some cases, a lack of interest. Instances of feeling irritated were expressed, such as “Especially with low-frequency sounds from strong winds, it can directly impact the human nervous system, causing headaches, nausea, and irritability” and “Feeling increasingly irritable, often tempted to get angry, and sometimes not in the mood to talk”. These examples highlight how the dust storm disrupts residents, causing disturbances in their daily lives. Additionally, another aspect of mental disturbance potentially linked to the dust storm is depression and fear. Examples include

“*Low visibility on the road almost led to a collision with another car, causing a moment of palpitations and fear” and “Furthermore, during a dust storm, the low visibility and dim light create a sense of oppression and fear*”.

These instances underscore the importance of studying this condition to mitigate its impact on the public’s well-being. Other responses indicated feelings of anxiety and a lack of interest, both of which could be attributed to the observable difficulties in daily life routines. 

The second theme involves the impact caused by past experiences, warranting attention from the public. Some participants shared that their impression of the dust storm is biased and dramatized by past experiences, making them more susceptible to mental impact. Noticeably, since individuals could have past experiences with dust storms, this could induce a distinctive pattern of responses. In one response, the participant mentioned that past experiences or impressions about past events could exaggerate the current psychological effect and make them vulnerable to the impact. Conversely, another participant reported that past experiences could have numbed their feelings about the current event. For example, the participant said, “Experiencing numerous dust storms has made me accustomed to them”. This distinctive pattern of responses to past severe weather conditions requires future research.

The final theme discerned in this question delves into the repercussions arising from apprehensions regarding personal or others’ health outcomes. It is noteworthy that this aspect has already surfaced in earlier phases of the study, demonstrating its recurrent presence and underscoring its significance within the research context. This concern about health conditions usually entails feelings of anxiety. For example, some participants might express concern about their children’s health and other family members: “Certainly, taking children outside during dust storms can be challenging, and prolonged exposure to such conditions may indeed have an impact on their well-being and development”. Likewise, this severe weather condition could also pose a significant threat to one’s personal health. For instance, “Long-term exposure to inhaling dust can indeed have a significant impact on the lungs. It may contribute to respiratory issues and other health concerns”. Overall, the mental impact induced by the dust storm, regardless of its source or attribution, appears to be significant, capturing a considerable proportion of the public’s attention. This observation underscores the need for more meticulous consideration and examination of its effects.

### 3.4. Incidents Related to Dust Storms That Have Affected Mental Well-Being

The following question complements the previous one, seeking to understand the circumstances under which people were more susceptible to the emotional impact caused by the dust storm. By scrutinizing the responses, two significant themes emerged: outcomes due to lessened outdoor activities and outcomes due to unfamiliarity with the weather conditions. According to the result, 21 out of 29 responses in this section attributed the emotional impact to being unable to go out. Three discernable emotional impact topics are a lack of interest or depression, feeling ruffled, and feeling unable to breathe.

To begin with, many people reported that being unable to perform outdoor activities induced their depressed feelings and even reduced interest in other things. One major area of lack of interest is the reduced social networking for many people. For example, 

“*In normal circumstances, I am a very optimistic person, but during dust storms, I tend to keep to myself, feeling that everything lacks enthusiasm” and “Low mood during dust storms may lead to a decrease in social interactions*”.

However, whether it is the visual perception of the daunting dust storm or other factors that make people feel depressed and unwilling to socialize remains unknown and requires further investigation.

Another central emotional theme caused by the inability to perform outdoor activities is irritation, which is also the most common theme reported by participants (11 responses). This bad temper can be attributed to a range of reasons. Some could be related to the additional difficulties faced at work, especially if the work needs to be performed outdoors, making people more irritated. In other cases, this increased irritation is simply a result of being confined in the house. For example, one participant reported that “During dust storms, when confined indoors due to the weather, tensions may arise more easily as there is no outlet for going out”. While the outcome is known, the specific reason or the neural mechanism underlying this outcome remains undetected. The last emotional theme is the feeling of being unable to breathe due to the confinement. Many people reported that this confinement and the raging yellow dust outside their window could induce a feeling of suffocation. Examples of responses include “Poor air quality can make one feel stifled or suffocated emotionally” and “Not refreshing or breathable enough”. This emotional feeling of suffocation is an emerging term that portrays participants’ emotions during the dust storm.

Besides these three emotional themes due to being unable to perform outdoor activities, other emotional changes could also be caused by unfamiliarity with the dust storm, including the duration of the condition, its source, and, more importantly, its outcome on one’s health. The underlying mechanism of this situation simulates that of generalized anxiety disorder and its link to uncertainty [21]. As one participant shared in the response, “Experiencing dust storm weather for the first time can indeed affect one’s mood. It is common for me to look for articles online about minimizing exposure and defense against dust storms”. Therefore, publicizing the information about dust storms to make more people aware of their outcomes becomes necessary and practical. 

### 3.5. Changes in Behaviors and Interaction Patterns during or after Dust Storms

The following questions function to locate more specific aspects of dust storm-induced behavioral changes. Three primary themes from the responses indicate reduced social interactions, heightened complaints and impatience, and decreased outdoor activities. According to the results, eight participants actively reported a tendency to reduce interactions with others during dust storms, while another six noted a general reduction in outdoor activities. Examples of responses include statements like 

“*During dust storms, there’s an unwillingness to go out and socialize. With improved weather conditions, there’s a greater willingness to embrace nature” and “Not wanting to go out is affecting both work and social interactions*”.

The observed pattern of reduced interaction may be associated with object-relations theory, which will be explored further in the subsequent section. Additionally, participants mentioned feeling less patient and more inclined to complain during severe weather conditions. This behavioral change can be categorized into two distinctive pathways: directly driven by the event or not. It could be considered directly driven by the dust storm when the participant mentioned that the aversive weather induces their changes. In contrast, circumstantial impacts were more likely delineated as unease or difficulties from work due to the aversive weather conditions that caused participants’ behavioral change. In addition to these common themes, one person expressed that the weather condition motivates him to engage in more environmental conservation actions. Therefore, the underlying mechanisms driving this adaptive behavior remain unknown, warranting future studies. 

### 3.6. Coping Strategies for Psychological Effects of Dust Storms

After thoroughly examining the behavioral and psychological impacts of dust storms on participants, another primary purpose of the study is to explore how residents in Beijing prefer to cope with this severe weather condition and seek insights into better assisting people living there. The study reveals numerous coping strategies residents employ, some unique to each participant’s lifestyle. This diverse range of coping strategies highlights the proactive nature of people living in Beijing and their eagerness to address this condition. Three major themes identified from the responses are acceptance and maintaining regular routines, diverting one’s attention, and reducing outdoor activities.

Diverting one’s attention emerges as the most common coping strategy. However, while 13 participants mention diverting their attention to alleviate the psychological impact of dust storms, their specific strategies vary widely. Only three residents mentioned listening to music and engaging in sports or other relaxation exercises like yoga as their diversion methods. The remaining residents employ unique coping methods, even if some of which may not be necessarily healthy or adaptive. These coping techniques include smoking, working, eating, playing games, changing cognition, watching TV shows, and drinking tea.

Acceptance is another common technique practiced by many residents in Beijing during dust storms. The prevalence of this theme can be attributed to the traditional Chinese belief in Daoism, which emphasizes flowing with nature. The specific benefits of this coping technique for indigenous residents could be studied further in the future. In addition to these three strategies, many people also prefer to engage in protective behaviors such as wearing masks and seeking relevant information to cope with the effects and soothe themselves. These protective measures contribute to a comprehensive approach to managing the impact of dust storms on daily life in Beijing. 

### 3.7. Suggestions for Governmental Support and Resources to Alleviate Effects

The final question in this survey aims to explore residents’ ideas for improving the current dust storm conditions, encompassing personal, community, and national levels. Except for two residents who lacked ideas about changing the condition, all participants expressed thoughts on improving the situation. Ideas include enhancing environmental protection awareness by promoting the negative impacts of dust storms, increasing vegetation, improving technology, opening public psychological support, implementing policy adjustments, and enhancing indoor activity facilities. The most prevalent suggestions are enhancing environmental protection awareness, which can be implemented at the community level, and increasing vegetation, which requires national-level efforts. 

## 4. Discussion

This qualitative research offers a detailed exploration of the ramifications of dust storms on the lives of Beijing residents, encompassing various facets such as behavioral patterns, psychological effects, and proposed solutions. Participants vividly recounted their experiences during the impactful dust storm of 2023, underscoring the urgency for policy reforms to safeguard the well-being of Beijing residents. The incorporation of responses from a diverse array of occupations ensures the comprehensiveness of the voices being heard. Notably, respondents span a broad occupational spectrum, encompassing indoor and outdoor workers, medical professionals, and individuals without specific expertise. This multidimensional study design aims to offer the public an extensive panorama of perspectives from every corner of society, intending to enhance the efficacy of future policy-making efforts.

Some remarkable sentiments toward dust storms were characterized by metaphorical expression like “end of the world”, stressing the severity of this weather phenomenon. Participants identified significant effects, including sensory impacts, emotional reactions, and disruptions to their daily routines. Visual perceptions emerged as a predominant factor influencing mental well-being, prompting a deeper exploration into specific impacts experienced by residents.

Significant disruptions to daily life were identified, primarily stemming from confinement and reluctance to engage in outdoor activities due to adverse weather conditions and associated health concerns. Emotional changes induced by the dust storm were also examined in detail, unveiling a nuanced spectrum of feelings and reactions among participants. In essence, this qualitative survey serves as a valuable resource for understanding the multifaceted impact of dust storms on the lives of Beijing residents. It offers insights that extend beyond immediate experiences to encompass broader behavioral, psychological, and societal dimensions, thereby contributing to informed decision-making and policy formulation aimed at mitigating the effects of dust storms in the future.

The primary focus of this study is the considerable impact that dust storms can have on mental health. This adverse impact can be categorized into direct effects of the weather conditions, effects stemming from past experiences, and effects arising from concerns about personal or others’ health conditions. Notably, past experiences can lead to different reactions to severe weather conditions, with some finding them more daunting and others accepting them based on prior encounters. Further research is needed to explore this distinction.

Addressing mental disturbances caused by unfamiliarity with dust storms should extend beyond individual experiences and become a communal or national concern. This collaborative effort aims to mitigate psychological distress associated with this climatic phenomenon. In addition to classification based on sources, unpleasant experiences can also be categorized according to circumstances: those caused by confinement and those not. Three prevalent themes at this juncture include a lack of motivation, heightened irritation, and a sensation of being unable to breathe or suffocation. These three themes in the current study concur with those from previous studies [13,14,15]. Self-isolation, as reported by many participants in the study during dust storms, has been found to induce varying levels of psychological distress. 

The potential attribution of these psychological impacts might resonate with object-relations theory [22], which posits a projection of human relationships with caregivers or internalized relationships with the ecosystem. During a dust storm, enforced home confinement and isolation from the ecosystem, represented by the swirling yellow dust, might lead to a disconnection with nature. This loss of connection could trigger negative feelings and even mental disturbances, aligning with the global trend of an increase in mental health issues. If this holds, targeted ecotherapy becomes imperative to foster a positive and adaptive relationship between individuals and nature [23]. An emerging theme in the responses is the feeling of suffocation, encompassing psychological and physiological well-being. This aspect warrants circumspect examination in subsequent studies to understand how the interplay between the body and mind contributes to this sensation. This unique feeling that appeared from the responses of Beijing residents also seems to resonate with the “Chinese Somatization” theory, which portrays various manifestations of depression symptoms cross-culturally [24]. More specifically, Chinese individuals have been observed to express somatic experiences rather than give direct descriptions of their psychological state when asked about their mental well-being. This unique conceptualization of mental health could be attributed to cultural factors and might demand further explication. On one hand, this distinctive sensation may be confined to Chinese or Asian samples, making it challenging to extrapolate to a broader population. On the other hand, it emphasizes the cultural specificity inherent in the Asian population, prompting further explorations and investigations within the Chinese sample.

While the primary focus of the current study during its design stage does not center on the socioeconomic impact of the dust storm event, the monetary hazards stemming from it must be considered. In economic terms, a natural disaster is conceptualized by Hallegatte and Przyluski [25] (p. 2) as “a natural event that causes a perturbation to the functioning of the economic system”. The economic impact, contingent on the nature of the cost, can be categorized into two distinct types: those directly induced by the event and those indirectly imposed. Both forms of impact may manifest due to the dust storm, necessitating thorough examination.

The direct impact pertains to the more immediate costs induced by the event, typically of short duration and occurring concurrently. Conversely, the indirect implications refer to longer-term situations, analogous to the destruction of infrastructure and constructions resulting from earthquakes or hurricanes, requiring several years for restoration and rebuilding [26]. In the context of the dust storm in Beijing, the indirect impact likely manifests as a general disruption to the economy, particularly the delay in production due to traffic closure. A secondary analysis conducted in 2000, examining the economic hazards of dust storms in Beijing, revealed that the delayed effects, i.e., the supply effect, hold a relatively superior position compared to the immediate impact [27]. 

The potentially more remarkable impact of the current study lies in the direct repercussions from two perspectives. Firstly, the most straightforward economic impact of the dust storm is attributed to absenteeism from work, a prevalent outcome noted in the survey responses. Widespread absenteeism at societal and national levels could contribute to an overall reduction in production. If this situation persists, it would mimic the indirect, long-term indirect effects of the dust storm. However, absenteeism’s tangible and concrete outcome reveals more obscured and clandestine outcomes. Given the prevalent mental disturbances in the study, a subsequent outcome could be a significant reduction in workplace productivity.

Girardi and her colleagues analyzed the implications of emotions and productivity in the workplace across five Dutch software development companies, finding a positive correlation between positive emotional valence and the productivity of software engineers [28]. They also identified that social and individual breaks facilitate and restore positive emotions at work. If this holds true, the depressed mood and reluctance to work expressed by many participants could severely undermine their motivation for work, resulting in economic losses for companies and reduced societal level. The beneficial impact of social breaks on emotional well-being also agrees with our findings about the emotional implications of confinement due to lack of socialization during dust storms.

Furthermore, another study [29] investigated the influence of emotion on productivity in the software engineering industry, partially attributed to the current rapid technological advancement and the desperate need for technical expertise. Besides the general correlation between positive emotion and enhanced creativity and performance, the study highlighted the contagious nature of human emotions, suggesting that negative emotions could potentially worsen outcomes. Given that many projects assigned to software engineers are preferably completed as a team, the emergence of negative emotion can alter behavior, initiating a chain reaction. These negative emotions, such as the tantrums and impatience observed in the study, could soon contaminate the entire group through interaction, leading to reduced productivity and performance and ultimately causing significant economic losses. 

Concerning coping mechanisms, diverting attention emerges as a major strategy, albeit with personalized variations among respondents. The uniqueness of these specific techniques hinders generalizability, but it aligns with the notion that individuals are best suited to determine what works for them. A notable theme is the acceptance and maintenance of regular routines, which echoes the principles of Acceptance and Commitment Therapy (ACT). ACT emphasizes acknowledging and gradually adjusting cognition to cope with inevitable challenges [30]. However, the external validity of these strategies is still waiting for proof, given the study’s confined geographical scope, focused solely within Beijing.

Participants in the study also expressed their views on improving the current dust storm conditions, highlighting several key themes: enhancing environmental protection awareness, increasing vegetation, and advancing technologies like reducing high-pollution factories. Notably, some participants recalled recent improvements potentially linked to the Beijing–Tianjin Dust Source Control Program since 2001. An assessment has revealed a positive trend in the annual Normalized Difference Vegetation Index (NDVI) across a significant portion of the total area, particularly in the dust source regions. This improvement, especially noticeable in spring when dust storms often occur, suggests an optimistic outlook for mitigating Beijing’s dust storm impacts [31]. However, community efforts to raise awareness about the impact of dust storms and promote environmental protection remain crucial.

## 5. Conclusions

While the current study provides a comprehensive exploration, several limitations should be acknowledged. Firstly, the sample size (n = 29) is limited. Future replications should prioritize a more extensive and diverse sample to enhance the robustness of the findings and improve generalizability. Moreover, since this study specifically investigates the impact of dust storms on Beijing residents, the applicability of the results to other regions or nations with distinct cultural backgrounds may be constrained.

Another limitation lies in the retrospective nature of all responses, as participants recalled experiences from dust storms that occurred in the past years. This raises concerns about response accuracy, given the chance that media might have biased participants’ memories. Future studies could address this by collecting responses in real-time or shortly after the dust storm event to ensure more precise data capture. Furthermore, as this study is qualitative in design, the current study lacks extensive quantitative inquiries and analyses, which could have explored the previously mentioned gender differences and variations in outcomes among participants with and without pre-existing health conditions. Future studies could benefit from incorporating more quantitative approaches to complement qualitative insights. 

Lastly, the bilingual survey employed back translation and was conducted by a single individual due to the study’s purpose. In subsequent studies, involving two or more translators for back translation could enhance interrater reliability and thereby ensure the translation accuracy across languages. This approach would contribute to maintaining the integrity and validity of the research findings.

While the socioeconomic ramifications of dust storms have been deliberately discussed in broader terms, and some correlations could be assumed from previous analyses, the current studies lack a more direct examination of this impact. Future research should focus on assessing the socioeconomic impact of natural hazards like dust storms more thoroughly [25]. If socioeconomic influence is a major concern for future studies, researchers could include more quantitative scales to measure both emotional fluctuation and the monetary loss incurred by companies. An example of the emotional assessment is the Positive and Negative Affect Schedule (PANAS), which could provide a detailed record of emotional fluctuation during dust storms. These data could later be correlated with financial data on monetary loss to capture a more systematic correlation between these variables. Such an approach would offer a more comprehensive understanding of how dust storms affect emotional well-being and lead to economic consequences, thereby facilitating a nuanced and informed analysis of their overall socioeconomic impact.

In conclusion, the primary objective of the current study is to comprehensively investigate the impact of dust storms on residents from both physical and mental perspectives. The findings highlight the vulnerability of residents’ mental well-being to the profound effects of dust storms, whether direct or indirect in nature. This study goes beyond merely documenting the impact by actively soliciting and collecting residents’ responses and thoughts on mitigating these effects.

A vital finding from the present study underscores a noticeable disparity between governmental policies and the actual needs of local residents. For example, despite ambitious afforestation programs like the “Three Norths Forest Shelterbelt” aimed at combating desertification and dust storms since 1970, their efficacy has recently been found to be overstated [32]. Policymakers must listen more closely to local voices to enact policy changes that better address residents’ real needs and enhance existing policies. Recommendations for improvement involve implementing targeted policy support, creating additional indoor facilities, and promoting environmental protection awareness on a broader scale. Implementing these changes not only carries the promise of ushering in a profound transformation in the lives of Beijing residents but also paints an optimistic prospect for the overarching well-being of the entire community.

## Data Availability

The raw data supporting the conclusions of this article will be made available by the authors on request.

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
