# Peer review of "The Effects of Dust Storms on People Living in Beijing: A Qualitative Study"

_ijerph, 2024, doi:10.3390/ijerph21070835_

Round 1

Reviewer 1 Report

Comments and Suggestions for Authors

I think this is an interesting study of an aspect of dust storms (not sandstorms) that has received little attention. With some adjustments, it will be a valuable contribution to the literature.

Title and throughout: This paper is about dust storms rather than sandstorms. The storms that reach Beijing arise far away, as the authors correctly state, and hence the small particles are dust-sized, not sand-sized, by the time they reach Beijing.

1.2 Health Hazards Directly Associated with Sandstorms: this review section is thin and should include more references (e.g. see Middleton, N., 2020. Health in dust belt cities and beyond—an essay by Nick Middleton. bmj371)

L50-59 the authors should read and cite this reference: Jones, B.A., 2023. Dust storms and human well-being. Resource and Energy Economics72, p.101362.

L109-110 there is a problem with the tense here. It should be past tense, not future: The sample will consist of Beijing residents who have experienced sandstorms and are currently residing or have lived in the city. The anticipated sample size…Participants will be recruited…

2. Materials and Methods: all seven questions should be included, either in the paper or an appendix.

2.2 Analysis: is it appropriate to use statistics on such a small sample size? In the conclusion, the authors suggest not, so I would delete.

L179 how can ‘a recent 2023 paper discussing increased sandstorm frequency in China’ (which is not referenced) ‘align’ with experience of storms in just a single year?

L569 ‘Another limitation lies in the retrospective nature of all responses, considering the event took place last year.’ So were the questions about a single event? Earlier in the paper, it seemed that the questions were about sandstorms in general.

L601 ‘a noticeable disparity between governmental policies and the actual needs of local residents’ what are the governmental policies mentioned here?

Comments on the Quality of English Language

Overall, pretty good although careful attention should be paid to tenses used.

Author Response

Comment 1: Title and throughout: This paper is about dust storms rather than sandstorms. The storms that reach Beijing arise far away, as the authors correctly state, and hence the small particles are dust-sized, not sand-sized, by the time they reach Beijing.

Response 1: Thank you for pointing this out. We agree with this comment. Therefore, we have replaced all sandstorm-related term with the dust storm. You could find this change in the introduction and throughout the entire paper.

Comment 2: Health Hazards Directly Associated with Sandstorms: this review section is thin and should include more references (e.g. see Middleton, N., 2020. Health in dust belt cities and beyond—an essay by Nick Middleton. bmj371)

 Response 2: We agree with this suggestion and add the citation in L48-49.

Middleton, N. (2020). Health in dust belt cities and beyond—an essay by Nick Middleton. bmj371.http://dx.doi.org/10.1136/bmj.m3089

Comment 3: L50-59 the authors should read and cite this reference: Jones, B.A., 2023. Dust storms and human well-being. Resource and Energy Economics72, p.101362.

Response 3: Same as above. This change can be found in L53-54 and in the reference section.

Jones, B. A. (2023). Dust storms and human well-being. Resource and Energy Economics72, 101362.

https://doi.org/10.1016/j.reseneeco.2023.101362

Comment 4: L109-110 there is a problem with the tense here. It should be past tense, not future: The sample will consist of Beijing residents who have experienced sandstorms and are currently residing or have lived in the city. The anticipated sample size…Participants will be recruited…

Response 4: Agree. The revised sentence can be found in L116. “The sample consisted of Beijing residents who had experienced sandstorms and were still residing or living in the city.”

Comment 5: Materials and Methods: all seven questions should be included, either in the paper or an appendix.

Response 5: Thank you for pointing this out. The full list of qualitative question can now be found in the appendix A.

- Could you briefly recall your recent experience of living in Beijing during sandstorms?

- In what ways do sandstorms impact your daily life and routines?

- Can you describe any changes in your mood or emotional well-being during sandstorms? Use words that best capture your emotions during these events.

- Can you tell me about any specific incidents or situations related to sandstorms that you think may have affected your mental well-being?

- Have you observed changes in your behavior or interactions with others during or after sandstorms? If yes, please elaborate on how these changes may have influenced your emotions.

- How do you cope with the psychological effects of sandstorms? Are there any specific strategies or activities you find helpful?

- What support or resources, if any, do you believe would be helpful in managing the mental health effects of sandstorms in Beijing? (This could be from personal, community, or the city in general).

Comment 6: Analysis: is it appropriate to use statistics on such a small sample size? In the conclusion, the authors suggest not, so I would delete.

Response 6: Agree. I removed all quantitative analysis in this paper since both reviewers have mentioned this question. Changes can be found throughout the paper.

Comment 7: L179 how can ‘a recent 2023 paper discussing increased sandstorm frequency in China’ (which is not referenced) ‘align’ with experience of storms in just a single year?

Response 7: Thank you for the comment. I have added the reference for this paper. Although this paper was published in 2023, it collected abundant of up-to-date information about sandstorm occurrences happened before 2023.

The revised sentence is like below. “This present finding corresponds with the ascending trend of dust storm frequency in China, reported earlier this year, as a reflection of climate change on the Mongolian plateau.”

Comment 8: L569 ‘Another limitation lies in the retrospective nature of all responses, considering the event took place last year.’ So were the questions about a single event? Earlier in the paper, it seemed that the questions were about sandstorms in general.

Response 8: Thank you for pointing this out. This question is not about a single event but sandstorms in general. The revised sentence can now be found in L583-585.  

Comment 9: L601 ‘a noticeable disparity between governmental policies and the actual needs of local residents’ what are the governmental policies mentioned here?

Response 9: Agree. Therefore, I did some researches and cited a prominent example in the paper. The change can be found in L617-620.

Wang, X. M., Zhang, C. X., Hasi, E., & Dong, Z. B. (2010). Has the Three Norths Forest Shelterbelt Program solved the desertification and dust storm problems in arid and semiarid China?. Journal of Arid Environments74(1), 13-22.https://doi.org/10.1016/j.jaridenv.2009.08.001

Reviewer 2 Report

Comments and Suggestions for Authors

The Introduction should likely not have sub-headings, and could be shortened to the most important aspects relating to the topic. 

 'Seven qualitative questions have been incorporated to prevent participant fatigue  is not a good justification  - ones doesn't include qualitative questions to prevent participants from getting tired or bored.  Quantitative  questions (eg line 105) should be rephrased to read 'demographic' questions.

Grounded theory analysis and thematic analysis are two different types of qualitative analysis techniques.  I would recommend leaving out grounded theory analysis and sticking with thematic content analysis.

The use of purposed and snowball sampling should be better justified, elaborated on.

The sample size is too small for any form of meaningful statistical analysis, and this should be cut out completely.  The study should be strictly qualitative in nature (and indeed the title of the study states this), with biographical information included.  There is more than sufficient qualitative findings to ground the study and paper in.

Literature cannot be cited in the findings (eg line 182) - this must be restricted to the Discussion section.

The paper should be streamlined so that the most succinct findings are reported.

Comments on the Quality of English Language

Extensive professional English language editing is required

Author Response

Comment 1: The Introduction should likely not have sub-headings, and could be shortened to the most important aspects relating to the topic. 

Response 1: Thank you for pointing this out. However, we think sub-sections and headings are closely related and correspond to the format of the later passage. So, we did not change this.

Comment 2: 'Seven qualitative questions have been incorporated to prevent participant fatigue  is not a good justification  - ones doesn't include qualitative questions to prevent participants from getting tired or bored.  Quantitative  questions (eg line 105) should be rephrased to read 'demographic' questions.

Response 2: Thank you for the suggestion. We replaced the term quantitative with demographic. The change can be found throughout the entire paper. Also, we have removed the sentence regarding “preventing participants from getting bored.” The change can be found in L108.

Comment 3: Grounded theory analysis and thematic analysis are two different types of qualitative analysis techniques.  I would recommend leaving out grounded theory analysis and sticking with thematic content analysis.

Response 3: Agree. According to the comment, we have removed the grounded theory analysis and stick with thematic content analysis.

Comment 4: The use of purposed and snowball sampling should be better justified, elaborated on.

Response 4: Thank you for pointing this out. We have provided more relevant justification for snowball sampling. The change can be found in L122-127.

Parker, C., Scott, S., & Geddes, A. (2019). Snowball sampling. SAGE research methods foundations.http://dx.doi.org/10.4135/

Comment 5: The sample size is too small for any form of meaningful statistical analysis, and this should be cut out completely.  The study should be strictly qualitative in nature (and indeed the title of the study states this), with biographical information included.  There is more than sufficient qualitative findings to ground the study and paper in.

Response 5: Thank you for pointing this out. We have removed all quantitative analysis from the paper. The change can be found throughout the paper. The change can be found in L194-197.

Comment 6: Literature cannot be cited in the findings (eg line 182) - this must be restricted to the Discussion section.

Response 6: Agree. Therefore, the literature cited in the result section was removed.

Comment 7: The paper should be streamlined so that the most succinct findings are reported.

Response 7: Thank you for pointing this out. Quantitative result has been removed from the present paper and only the more relevant qualitative analysis was kept.  

Round 2

Reviewer 2 Report

Comments and Suggestions for Authors

- Abstract - remove the 'headings' (eg (1) Background, so that it compiles to the journal requirements.

- Line 7, rephrase to read: Dust storms, which are a common aversive occurrences in northern China

- I still recommend the removal of sub-titles in the Introduction, this can be deferred to and editorial decision though.

- Line 37, again amend sentence construction to read: Dust storms, which are typical aversive events in northern China

- Line 105, rephrase to read: The present study adopted a qualitative research design, utilizing an online survey

- I am not sure is a survey and qualitative research is compatible?  The researchers should clarify here.

- you state that both purposive and snowballing recruitment methods was used, however, only the snowballing approached is discussed. It seems like purposive must be left out (line 118)

- as this is a qualitative study, this section should be removed (lines 145-151): 

2.2 Analysis

Given that the data and responses primarily consist of written responses collected 146 through a Qualtrics survey, the analysis for the current study was predominantly 147 conducted using NVivo 14.23.0. This software is specifically designed for the analysis of 148 qualitative and mixed-method research, adept at handling unstructured text, audio, video, 149 and image data from various sources, including interviews, focus groups, surveys, social 150 media, and journal articles [20].

- line 164, rather use the term 'study' as opposed to 'survey'

- line 169, remove additional fullstop

- line 174 which will be discussed 173 later in the article

Though much improved,  the article still requires the input of a professional English language editor - Ive highlighted some of the language corrections above, but will not do so for the entire article. 

Conlusion - line 537, what is meant by this? Elaborate or correct  primarily conducted for a school project

Comments on the Quality of English Language

The article has been significantly improved.  Corrections (minor) are still required, however.

Author Response

Comment 1: Abstract - remove the 'headings' (eg (1) Background, so that it compiles to the journal requirements.

Response 1: Agree. All headings in the introduction were removed. Changes can be found in the introduction and throughout the entire paper.

Comment 2: Line 7, rephrase to read: Dust storms, which are a common aversive occurrences in northern China

Response 2: Thank you for pointing this out. The sentence were rephrased. The change can be found in L7. Additional attention to the grammatical structure of the passage would also be paid.

Comment 3: I still recommend the removal of sub-titles in the Introduction, this can be deferred to and editorial decision though.

Response 3: Agree. Sub-titles in the introduction were removed. Changes can be found in the introduction.

Comment 4: Line 37, again amend sentence construction to read: Dust storms, which are typical aversive events in northern China

Response 4: Thank you for pointing this out. The sentence were rephrased. The change can be found in L37.

Comment 5: Line 105, rephrase to read: The present study adopted a qualitative research design, utilizing an online survey

Response 5: Agree. The sentence were rephrased. The change can be found in L106.

Comment 6: I am not sure is a survey and qualitative research is compatible?  The researchers should clarify here.

Response 6: The survey is the tool or measure to collect participant’s responses, which comprised of primarily open-ended question. All written questions later went through qualitative analysis, but I believe they are compatible.

Comment 7: you state that both purposive and snowballing recruitment methods was used, however, only the snowballing approached is discussed. It seems like purposive must be left out (line 118)

Response 7: Thank you for pointing this out. The term purposive was removed from the paper. Change can be found in L121.

Comment 8: as this is a qualitative study, this section should be removed (lines 145-151): 

Response 8: Thank you for pointing this out. The entire section had been removed from the passage. Change can be found in L149.  

2.2 Analysis

Given that the data and responses primarily consist of written responses collected 146 through a Qualtrics survey, the analysis for the current study was predominantly 147 conducted using NVivo 14.23.0. This software is specifically designed for the analysis of 148 qualitative and mixed-method research, adept at handling unstructured text, audio, video, 149 and image data from various sources, including interviews, focus groups, surveys, social 150 media, and journal articles [20].

Comment 9: line 164, rather use the term 'study' as opposed to 'survey'

Response 9: Agree. The change can be found in L166.

Comment 10: line 169, remove additional fullstop

Response 10: I do not quite understand which full stop you were referring to since L170 – L187 had been removed. Could you please elaborate on this?

Comment 11: line 174 which will be discussed 173 later in the article

Response 11: L170-187 has been completely removed from the paper (see Result section), so I am not quite clear about this comment.

Comment 12: Though much improved,  the article still requires the input of a professional English language editor – I’ve highlighted some of the language corrections above, but will not do so for the entire article. 

Response 12: Thank you for pointing this out. The entire paper had went through careful grammatical inspection. Change can be found throughout the paper.

Comment 13:  line 537, what is meant by this? Elaborate or correct  primarily conducted for a school project

Response 13: Thank you for pointing this out. “Primality conducted for a school project” was removed. Change can be found in L587.